# Leaf Transcriptome Analysis of Broomcorn Millet Uncovers Key Genes and Pathways in Response to *Sporisorium destruens*

**DOI:** 10.3390/ijms22179542

**Published:** 2021-09-02

**Authors:** Fei Jin, Jiajia Liu, Enguo Wu, Pu Yang, Jinfeng Gao, Xiaoli Gao, Baili Feng

**Affiliations:** State Key Laboratory of Crop Stress Biology in Arid Areas, College of Agronomy, Northwest A&F University, Yangling 712100, China; 18833033167@163.com (F.J.); ljjzl2014@163.com (J.L.); WEG2990762254@163.com (E.W.); yangpu5532@hotmail.com (P.Y.); gaojf7604@126.com (J.G.); gao2123@nwsuaf.edu.cn (X.G.)

**Keywords:** broomcorn millet, stress, *Sporisorium destruens*, transcriptome, *Panicum miliaceum*, smut

## Abstract

Broomcorn millet (*Panicum miliaceum* L.) affected by smut (caused by the pathogen *Sporisorium destruens*) has reduced production yields and quality. Determining the tolerance of broomcorn millet varieties is essential for smut control. This study focuses on the differences in the phenotypes, physiological characteristics, and transcriptomes of resistant and susceptible broomcorn millet varieties under *Sporisorium destruens* stress. In diseased broomcorn millet, the plant height and stem diameter were reduced, while the number of nodes increased. After infection, the activities of superoxide dismutase and peroxidase decreased, and malondialdehyde and relative chlorophyll content (SPAD) decreased. Transcriptome analysis showed 514 and 5452 differentially expressed genes (DEGs) in the resistant and susceptible varieties, respectively. The Kyoto Encyclopedia of Genes and Genomes (KEGG) enrichment analysis of DEGs showed that pathways related to plant disease resistance, such as phenylpropanoid biosynthesis, plant–pathogen interaction, and plant hormone signal transduction, were significantly enriched. In addition, the transcriptome changes of cluster leaves and normal leaves in diseased broomcorn millet were analysed. Gene ontology and KEGG enrichment analyses indicated that photosynthesis played an important role in both varieties. These findings lay a foundation for future research on the molecular mechanism of the interaction between broomcorn millet and *Sporisorium destruens*.

## 1. Introduction

Smut is a common plant disease [1,2], and is one of the major diseases of broomcorn millet (*Panicum miliaceum* L.), able to seriously affect production. This disease not only reduces yield, it also leads to the contamination of grains and straws [3]. The incidence is generally 5–10%, although in severe cases it may reach 40% [4]. Smut is a systemic invasive disease in which the pathogen infects the host at the seedling stage, but obvious symptoms appear at the heading stage and beyond [5]. The pathogens of smut include *Basidiomycota* species and *Sporisorium destruens*. After *Sporisorium destruens* infects broomcorn millet, it ruins the grain harvest, and pathogens attach to the soil or seed to overwinter and become a source of infection. A high-temperature, high-humidity environment is conducive to the invasion of broomcorn millet by *Sporisorium destruens* [6]. At the early heading stage, diseased broomcorn millet does not show significant differences compared to uninfected plants, and smut prevention and treatment in production is more difficult.

To prevent pathogen invasion, plants have evolved specific immune strategies, including pathogen-associated molecular pattern (PAMP)-triggered immunity (PTI) [7], which identifies pathogen invasion, followed by a host defence response [8,9]. Cytoplasmic Ca^2+^ concentration has been recognised as an essential signal for PTI [10]: during PTI, host cells perceive PAMP through pattern-recognition receptors; this induces an increase in the concentration of cytosolic Ca^2+^ and leads to activation of calcium-based defence responses [10]. Stomatal behaviour not only influences the balance between CO_2_ uptake and water loss, it also increases the risk of lethality upon stress; the hypersensitive response (HR) is a common feature of plant immune responses, and a type of programmed cell death [11,12]. Ca^2+^ is known to affect the HR, as well as cell wall and stomatal changes [11,13,14].

Broomcorn millet, an allotetraploid from China with a long cultivation history [15], is cultivated in Europe, the Middle East, and other regions [16], has high nutritional value, and can be used for food, feed, and medicine [17]. Broomcorn millet is tolerant to drought stress and barren soil, and can serve as a pioneer crop, suitable for sustainable agricultural practices [18]. Although the production of broomcorn millet has many advantages, it is grown on a small scale in only a few regions [19], and head smut threatens this industry by substantially reducing yields. Research on broomcorn millet smut has focused mainly on physiological and biochemical responses, agronomic traits, and the effect on yield [20,21,22], as well as fungicide screening and resistance identification [23,24]. Furthermore, there are few reports on the breeding of broomcorn millet smut resistance and the plant–pathogen interaction mechanisms that negatively impact on broomcorn millet production and smut research.

The high-throughput capabilities of RNA sequencing (RNA-Seq) technology can be used to evaluate gene transcription levels [25,26] and study the interaction mechanism between plants and pathogens at the molecular level [27]. In this study, we investigated and compared the expression of differentially expressed genes (DEGs) in susceptible and resistant broomcorn millet varieties under *Sporisorium destruens* stress. The changes in plant–pathogen interaction, oxidation-reduction process, and plant hormone signal transduction pathways were analysed. These results provide a valuable reference information for the study of interaction between broomcorn millet and *Sporisorium destruens*.

## 2. Results

### 2.1. Phenotype Analysis of Resistant/Susceptible Broomcorn Millet after Sporisorium destruens Infection

At the early heading stage (about 40 days after planting) of two broomcorn millet varieties, there was no significant difference between uninoculated controls (CK; R0 and S0) and *Sporisorium destruens*-inoculated samples (R1 and S1). After the heading stage, the ears at the top of the diseased plant were transformed into small cluster leaves, while other treatments appeared normal (Figure 1A). The plant height and stem diameter of diseased broomcorn millet of both varieties were significantly decreased (Figure 1B,C), and the node number increased (Figure 1D). The differences between varieties were not significant (*p* > 0.05), except for the stem thickness of uninoculated plants (R0 vs. S0; *p* < 0.05), and the node number of inoculated plants (R1 vs. S1; *p* < 0.05).

### 2.2. Redox Status and Relative Chlorophyll Content of Broomcorn Millet Leaves under Sporisorium destruens Stress

To reveal the physiological responses in the leaves of the two broomcorn millet varieties under *Sporisorium destruens* stress, we determined the activities of superoxide dismutase (SOD) and peroxidase (POD), as well as malondialdehyde (MDA) and relative chlorophyll content (SPAD) index (Figure 2). The differences in SOD activity, MDA content, and SPAD index showed similar trends, i.e., they were lower in the inoculated leaves than in the uninoculated leaves. In the smut-resistant variety, Bameng Xiaohei mi (BM), POD activity was lower in uninoculated leaves than in inoculated leaves, whereas in the smut-susceptible variety, Nianfeng No. 5 (NF), the POD activity in uninoculated leaves was higher than in inoculated leaves.

### 2.3. Data Quality and DEGs in Two Broomcorn Millet Varieties under Sporisorium destruens Stress

We obtained 15 complementary DNA (cDNA) libraries with high sequencing quality using RNA sequencing (RNA-Seq) (Appendix A). Principal component analysis (PCA) clusters samples based on the gene expression of samples, reflecting the repeatability of the samples. The R0-1 sample was identified as an outlier and removed in the following analysis (Figure 3A, Appendix A). Venn analysis indicated that there were 30,545 co-expressed genes among the samples; R0, R1, S0, and S1 had 370, 300, 336, and 1686 expressed genes, respectively (Figure 3B). In BM, 514 genes were defined as DEGs in R1 compared with R0, including 28 up-regulated and 486 down-regulated DEGs. In NF, 5452 genes were identified as DEGs, with 3989 up-regulated and 1463 down-regulated. A total of 231 genes were identified as DEGs in the four samples of the two varieties. Compared with CK, the inoculated treatments had higher levels of DEGs. Overall, more DEGs were characterised in NF than in BM (5452 vs. 514, respectively; 10.6-fold difference). In NF, most of the DEGs (3989 out of 5452; 73.17%) were up-regulated, and 1463 DEGs were down-regulated; however, only 28 DEGs (accounting for 5.4%) were characterised as being up-regulated in BM, and the rest (486 out of 514; 94.6%) were down-regulated (Figure 3C). This suggests that NF was inclined to activate gene expression to cope with *Sporisorium destruens* stress.

### 2.4. Gene Ontology (GO) Enrichment Analysis of DEGs

A hypergeometric distribution was used to divide the DEGs into GO terms. BM and NF had 350 and 2517 enriched GO terms, respectively. Specifically, in BM, GO enrichment analysis included 82 molecular functions (MFs), 100 biological processes (BPs), and 1 cell component (CC), whereas the GO enrichment analysis of NF consisted of 151 MFs, 331 BPs, and 42 CCs (Appendix A). Among the first twenty enriched GO terms in the two broomcorn millet varieties (Figure 4A), BM had only 12 terms that were significantly enriched (*p* < 0.05). Additionally, five terms were the same in the two varieties: oxidation-reduction process (GO: 0055114), oxidoreductase activity (GO: 0016491), catalytic activity (GO: 0003824), iron ion binding (GO: 0005506), and dioxygenase activity (GO: 0051213). In the term type classification, the oxidation-reduction process belongs to BP, and the remaining four items belong to MF, which showed that the MF of broomcorn millet was greatly affected after *Sporisorium destruens* inoculation. The catalytic activity was the term with the largest number of DEGs in both broomcorn millet varieties: compared with R0, 212 DEGs were enriched in R1; and 1412 DEGs were also enriched in S1 compared with S0.

### 2.5. KEGG Enrichment Analysis

The DEGs were mapped to the reference pathways in KEGG to investigate the interaction between *Sporisorium destruens* and broomcorn millet. KEGG enrichment analysis showed that 197 DEGs in R0 vs. R1, and 1286 DEGs in S0 vs. S1, were mapped onto 64 and 114 KEGG pathways, respectively, and there were 12 and 27 KEGG pathways that were significantly enriched, respectively (*p* < 0.05) (Appendix A). Figure 4B plots the top 20 enrichment pathways of two broomcorn millet varieties, indicating that four significant enrichment pathways appeared in both varieties at the same time, namely phenylpropanoid biosynthesis, plant–pathogen interaction, plant hormone signal transduction, and alpha-linolenic acid metabolism pathways. In addition, pathways related to photosynthesis, photosynthesis-antenna proteins, and porphyrin and chlorophyll metabolism were significantly enriched in S0 vs. S1 (*p* < 0.05).

### 2.6. DEGs Analysis of Different Leaf Types of Diseased Broomcorn Millet

To further explore the interaction mechanism between diseased broomcorn millet and *Sporisorium destruens*, the transcriptomes of normal leaves on diseased plants (S1) and top cluster leaves (S2) were analysed (Figure 5A). Compared with S1, there were 4213 DEGs in S2, including 1456 up-regulated DEGs and 2757 down-regulated DEGs (Figure 5B). The number of down-regulated DEGs in S2 was 1.9-fold higher than that of up-regulated DEGs, which was contrary to the trend of S0 vs. S1. Different mechanisms in S0 vs. S1 and S1 vs. S2 were inferred from the differences in the changes between susceptible varieties.

### 2.7. GO and KEGG Enrichment Analyses of S1 vs. S2

GO enrichment analysis showed that 3061 DEGs were distributed in 516 pathways in S1 vs. S2, of which 261 were significantly enriched (Appendix A). Of these 261 pathways, 48, 161, and 52 were CC BP and MF terms, respectively. In the first 20 pathways enriched by GO, there were 7 BPs, 11 CCs, and 2 MFs (Figure 5C). Notably, among the 20 terms, those relating to photosynthesis accounted for a large proportion, such as photosynthesis (GO: 0015979), light harvesting in photosystem I (GO: 0009768) in BP, photosystem (GO: 0009521) in CC, and chlorophyll binding (GO: 0016168) in MF. Combined with the comprehensive analysis of the GO enrichment results in S0 vs. S1, we confirmed that photosynthesis played an important role in the interaction between *Sporisorium destruens* and broomcorn millet.

Compared with S1, there were 1478 DEGs involved in the KEGG pathways in S2, and the results showed that these DEGs were distributed in 120 pathways, of which 33 pathways were significantly enriched (Appendix A). It was apparent that benzoic acid biosynthesis, photosynthesis, and photosynthesis-antenna pathways were the three most significantly enriched pathways (Figure 5D). Phenylpropane biosynthesis, photosynthesis, and plant–pathogen interaction pathways had the largest amount of enriched DEGs_._ KEGG enrichment analysis indicated that S0 vs. S1 showed different response mechanisms between *Sporisorium destruens* and broomcorn millet, although both involved photosynthesis.

### 2.8. DEGs Involved in Plant Hormone Signal Transduction Pathway

In plant hormone signal transduction, the jasmonate ZIM (JAz)-domain-containing protein family is a key component of the jasmonic acid (JA) signalling pathway [28]. In the JAz family, 19 genes in five samples of two broomcorn millets were defined as DEGs (Figure 6). DEGs in the two varieties showed different trends: in BM, compared with R0, seven DEGs were down-regulated in R1; in S0 vs. S1, 18 DEGs were defined as being up-regulated; the 9 DEGs annotated in the JAz family of S1 vs. S2 included one down-regulation and eight up-regulations. After *Sporisorium destruens* infection, the expression of JAz may be related to the resistance of broomcorn millet; in BM varieties, all the DEGs annotated in the JAz family were down-regulated, whereas in NF, most DEGs were up-regulated. There were three DEGs annotated in the GH3 auxin-responsive gene family: one DEG was defined as up-regulated in S0 vs. S1, and two DEGs were down-regulated in S1 vs. S2. Furthermore, the expression of 12 genes in the auxin-responsive protein (IAA) family had changed: there were four up-regulated DEGs and one down-regulated DEG in S0 vs. S1, and two up-regulated DEGs and five down-regulated DEGs in S1 vs. S2 (Appendix A). In the resistant variety of broomcorn millet, no DEGs were annotated in the IAA and GH3 families.

### 2.9. Analysis of Enriched DEGs in the Plant–Pathogen Interaction Pathway

Knowledge of the plant–pathogen interaction pathway is key to understanding the interactions between plants and pathogens. In the plant–pathogen interaction pathway of broomcorn millet and *Sporisorium destruens*, there were 13 DEGs in BM, 56 DEGs in S0 vs. S1, and 64 DEGs in S1 vs. S2, respectively (Appendix A). Interestingly, most of the DEGs in BM were down-regulated (11 out of 13), whereas in NF the up-regulated DEGs accounted for a larger proportion. Among the 56 DEGs enriched in S0 vs. S1, the number of up-regulated DEGs was 40. Furthermore, 43 DEGs were up-regulated in S1 vs. S2. These DEGs were mainly distributed in the calmodulin (CaM), calmodulin-like (CML), and 3-ketoacyl-CoA synthase (KCS) families. In addition, the expression of DEGs was affected in the calcium-dependent protein kinase (CDPK), respiratory burst oxidase homologue, and PTI families of NF, but this was not observed in BM (Figure 7A). DEGs enriched in plant–pathogen interaction pathways may affect cell wall reinforcement, stomatal closure, HR, defence-related gene induction, phytoalexin accumulation, miRNA production, suppression of plant HR, and defence responses. The leucine-rich repeat (LRR) transmembrane receptor kinase, flagellin-sensitive 2 (FLS2), is essential for flagellin perception [29]. There were three DEGs annotations in the FLS2 family, *Longmi038613* and *Longmi019029* were up-regulated in S0 vs. S1; *Longmi038613* and *Longmi019029* were down-regulated in S1 vs. S2. In this study, these DEGs may act as receptors or kinases in the HR, cell wall reinforcement, and stomatal closure.

### 2.10. Expression Correlation Analysis of Enriched DEGs in Plant–Pathogen Interaction Pathway

Expression correlation analysis is based on the correlation of gene expression and is used to identify key genes by analysing the connections between genes (Appendix A). A correlation analysis of 95 DEGs in the plant–pathogen interaction pathway was carried out, and the DEGs with strong correlation was selected to draw the correlation network diagram (Figure 7B–D). Analysis of the top 10 genes with strong correlation in each group showed that both *Longmi000712* and *Longmi036891* were strongly correlated in R0 vs. R1, and in S0 vs. S1; these two genes may be related to the smut resistance of broomcorn millet. In the two groups of susceptible broomcorn millet, *Longmi015394* and *Longmi023157* showed a strong correlation, which may be related to the pathogenic mechanism of broomcorn millet. In addition to the above-mentioned genes, there were 8 DEGs with strong correlation in each treatment. These DEGs may be the reasons for the different resistances and pathogenesis of broomcorn millet.

### 2.11. Real-Time PCR Verification of DEGs

Ten DEGs were selected from the transcriptome results for RT-PCR verification of their reliability. The primer sequence is shown in Appendix A. The RT-PCR verification and RNA-Seq data were roughly the same (Figure 8), indicating that the transcriptome results were reliable.

## 3. Discussion

Smut is a common crop disease which damages ears and grains and reduces economic output. Research on smut has mainly focused on crops such as rice, sugarcane, and maize [1,30,31]. This study used RNA-Seq to analyse the transcriptional changes in smut-resistant and smut-susceptible broomcorn millet varieties after inoculation with *Sporisorium destruens*. Sequencing analysis identified 514 and 5452 DEGs in the resistant and susceptible cultivars, respectively. This result was consistent with the study by Cheng et al. [32], in which inoculation of Tainung 67 (resistant rice cultivar) and Zerawchanica karatals (susceptible rice cultivar) with *Fusarium fujikuroi* resulted in the appearance of 118 and 169 DEGs, respectively. Hence, susceptible varieties tend to activate gene expression in response to stress.

Plant immunity is a complicated process affected by extracellular or intracellular receptors that recognise PAMPs or effectors in PTI and effector-triggered immunity systems [9,33]. The defence response mechanisms and role of receptors have hardly been studied in broomcorn millet, but they have been reported in plants such as *Arabidopsis*, tomato, and rice [29,34,35,36]. In the plant–pathogen interaction pathway, there were 13, 56, and 64 DEGs in R0 vs. R1, S0 vs. S1, and S1 vs. S2, respectively. Compared with BM, NF expressed more DEGs after inoculation, and these DEGs may play an important role in the interaction between broomcorn millet and *Sporisorium destruens*.

Ca^2+^ acts as a second messenger in stress response signalling pathways [37], involving CaM, CaM-like (CML) proteins, and CDPKs [38]. Cyclic nucleotide-gated channels (CNGCs) reportedly play an important role in plant immunity, plant hormones, and response to stressors [36,39]. The CDPKs are crucial sensors of changes in Ca^2+^ concentration and have multiple roles in the stress tolerance of plants [40,41], while CaM plays a crucial role in plant defence signalling [42]. After inoculation with *Sporisorium destruens*, among the annotated DEGs were one gene in the CNGC family, 12 in the CDPK family, and 23 in the CaM/CML family (Appendix A). In a study on canker disease in pitaya (*Hylocereus polyrhizus*) [43], infected tissue had two up-regulated unigenes and one down-regulated unigene in the CDPK family, and one up-regulated unigene annotated in the CaM/CML family. Herein, in the interaction between broomcorn millet and *Sporisorium destruens*, 12 annotated DEGs belonged to the CDPK family. Furthermore, eight DEGs in S0 vs. S1 were all up-regulated; 10 DEGs in S1 vs. S2, including eight up-regulated and two down-regulated. There were 23 annotated DEGs from the CaM/CML family, including two shared DEGs. Interestingly, the DEGs annotated in BM were all down-regulated, whereas all were up-regulated in NF. In the plant–pathogen interaction pathway, changes in the expression of these DEGs can affect cell wall reinforcement, stomatal closure, and HR, thereby affecting plant resistance.

In plants, disease resistance depends on the presence of complementary gene pairs in the host and pathogen. These are divided into resistance (R) genes and avirulence (Avr) genes, which can effectively curb the invasion of pathogens [44,45]. R genes share some conserved domains, according to the arrangements of their functional domains, which can be grouped into five classes [46], based on the presence of an N-terminal domain and a LRR domain. The LRR domain is responsible for specific pathogen recognition [47]. The resistance genes of the LRR domain have been reported in rice, wheat, and soybean [48,49,50], but not in broomcorn millet. Here, in the plant–pathogen interaction pathway, there were three annotated DEGs from the FLS2 family in the LRR domain, including one shared DEG: *Longmi038613* was defined as up-regulated DEG in S0 vs. S1, whereas it was defined as down-regulated DEG in S1 vs. S2 (Appendix A). Different samples had different expression levels of *Longmi038613*, which requires verification to facilitate future research.

JA is an important phytohormone that regulates the defence responses of a plant [51,52]. In *Arabidopsis*, JA signalling has been negatively and positively associated with resistance against *Fusarium graminearum* [53]. Among the plant hormone signalling pathways, 18 DEGs in NF were up-regulated in the JAz family, and seven DEGs in BM were identified as being down-regulated. These results were consistent with a study on the response of broomcorn millet to the stress of *Sporisorium destruens*; the JA content increased after broomcorn millet infection [21]. It was inferred that invasion by *Sporisorium destruens* increased the JA content in the leaves of diseased broomcorn millet to resist *Sporisorium destruens* invasion. However, Cheng’s research results showed that the genes associated with the JA biosynthetic process were up-regulated in the resistant cultivar, Selenio, and down-regulated in the susceptible Dorella cultivar [32], which was contrary to our result. Research on the role of JA in the interaction between broomcorn millet and smut needs further investigation.

When plant cells are subjected to adverse conditions, large amounts of reactive oxygen species (ROS) are produced [54], and the burst of ROS inhibits the process of chlorophyll electron transfer and photorespiration [55]. A high antioxidant capacity to scavenge the toxic ROS has been linked to stress tolerance [56]. The ROS in turn cause damage to lipids, and MDA content was taken as an indication of lipid peroxidation level [57]. Furthermore, changes in antioxidant protection enzymes have been reported in broomcorn millet [21,58]. Therefore, we measured SOD and POD activities in the leaves of each sample. Compared with CK, SOD activity decreased in both varieties after inoculation. In the transcriptome results, there were no genes annotated in the SOD family of BM varieties, but in S0 vs. S1, three DEGs were annotated in the SOD family, all of which were down-regulated (Appendix A). There was one down-regulated DEG annotation in S1 vs. S2. This trend was consistent with the measured SOD activity in NF leaves. In sugarcane leaves infected by orange rust, the SPAD index of reflecting chlorophyll content decreased with increasing rust rating [59]. In the photosynthetic report of *Ustilago maydis* infection of maize, the relative chlorophyll content of leaves was reduced after *Ustilago maydis* infection [60]. The SPAD index of broomcorn millet leaves after inoculation with *Sporisorium destruens* was lower than that of uninoculated controls; this may be the result of chlorophyll degradation by *Sporisorium destruens* infection. Furthermore, KEGG enrichment analysis indicated significantly enriched pathways related to photosynthesis, and chlorophyll metabolism. The role of photosynthesis in the interaction between broomcorn millet and *Sporisorium destruens* should be studied further.

## 4. Materials and Methods

### 4.1. Plant Material, Sporisorium destruens Inoculation and Sample Collection

The broomcorn millet varieties used in this study, namely Nianfeng No. 5 (NF) and BM, were provided by Northwest A&F University (Yangling, China). BM was considered a smut-resistant variety; NF was considered a smut-susceptible variety. A two-year field survey showed that the incidence of *Sporisorium destruens* infection in BM and NF was 4.59% and 47.5%, respectively. The spores of *Sporisorium destruens* were collected in the experimental field at Northwest A&F University, dried in the shade at room temperature, surface-disinfected with 75% ethanol, and placed on an ultra-clean workbench. Then, the *Sporisorium destruens* spores were cut and collected with a sterilised blade, and the samples were placed in a kraft paper bag in a refrigerator at 4 °C. It was removed from the refrigerator one day before use and kept at room temperature.

The experiment was carried out in a greenhouse at Northwest A&F University in 2020. The soil and substrate (Pindstrup, Co., Ltd. Shanghai, China) were mixed in a 1:1 ratio. Seeds inoculated by saturation inoculation were mixed with *Sporisorium destruens* spores in a 20:1 mass ratio, representing the inoculation treatment. Uninoculated seeds served as the experimental control (CK). The seeds were planted in a black flowerpot (10 plants/pot) with a diameter of 20 cm and a height of 10 cm, which were calculated based on the field density. The temperature and duration of illumination of the day/night cycles were set to 30 °C/18 °C and 14 h/10 h, respectively. Relative humidity was constant at 60%, and watering was performed every 1–2 days. The illumination intensity was set to 600 µmol m⁻^2^ s⁻^1^. After showing symptoms at the heading stage, the lower leaves of the ear were collected: R0 (BM-CK), R1 (BM-inoculated), S0 (NF-CK), and S1 (NF-inoculated). In addition, the top cluster leaves of NF-diseased plants were collected and named S2. Collected leaf samples were stored in ultra-low-temperature refrigerators for Illumina deep sequencing, antioxidant enzyme activity assays, and RT-PCR validation.

### 4.2. Phenotype Analysis of Broomcorn Millet

After broomcorn millet heading, the ears of the diseased plant turn into small cluster leaves. At this stage, the phenotype of each treatment was evaluated. Five replicates were used to determine the differences in mean plant height (cm), stem diameter (mm), and node numbers of control plants and infected plants. Plant height and stem thickness were measured manually using a ruler and a digital calliper, respectively.

### 4.3. Measurement of SOD Activity, POD Activity, MDA Content, and SPAD Index 

SOD and POD activities, as well as the MDA content, in the leaves under the ears of R0, R1, S0, S1, and S2, were determined using commercial detection kits (Solaibao Technology Co., Ltd., Beijing, China), according to the manufacturer’s instructions. The SPAD index was determined using a non-invasive, handheld meter (SPAD-502, Konica Minolta, Tokyo, Japan).

### 4.4. Transcriptome Sequencing of Broomcorn Millet Leaves

Total RNA was extracted from the tissue using Plant RNA Purification Reagent for plant tissue, according to the manufacturer’s instructions (Invitrogen, Carlsbad, CA, USA), and genomic DNA was removed using DNase I (Takara Bio, Kusatsu, Japan). Then RNA quality was determined using a 2100 Bioanalyzer (Agilent Technologies, Inc., Santa Clara, CA, USA) and quantified using the ND-2000 (NanoDrop Technologies, Inc., Wilmington, DE, USA).

The RNA-Seq library was prepared following the TruSeqTM RNA sample preparation kit from Illumina (San Diego, CA, USA) following the manufacturer’s instructions. Briefly, messenger RNA was isolated according to the polyA selection method using oligo (dT) beads and then fragmented using fragmentation buffer. Second, double-stranded cDNA was synthesised using a SuperScript double-stranded cDNA synthesis kit (Invitrogen) with random hexamer primers (Illumina). The double-stranded cDNA was purified and ligated to adaptors for Illumina paired-end sequencing. Libraries were selected for cDNA target fragments of 300 bp on 2% Low Range Ultra Agarose, followed by PCR amplification using Phusion DNA polymerase (New England Biolabs, Ipswich, MA, USA) for 15 PCR cycles. The resulting cDNA libraries were sequenced at Shanghai Majorbio Biopharm Technology Co., Ltd. (Shanghai, China) using the Illumina HiSeq xten/NovaSeq 6000 sequencing system.

### 4.5. Transcriptome Analysis

To identify the DEGs in broomcorn millets, transcript expression levels were calculated according to the transcripts per million reads method. RNA-Seq by Expectation-Maximization was used to quantify the gene abundance (http://deweylab.biostat.wisc.edu/rsem/ (accessed on December 2020)). Differential expression analysis was performed using DESeq2, and genes with |log2FC|≥1 and *p* ≤ 0.05 (DESeq2) were used as DEGs. Functional enrichment analysis of GO and KEGG was performed to identify the significant terms and pathways in terms of the Bonferroni-corrected *p*-value ≤ 0.05, compared with the reference genome (http://bigd.big.ac.cn/gwh/Assembly/131/show (accessed on November 2020)). GO enrichment terms and KEGG enrichment pathways were analysed using GOATOOLS (https://github.com/tanghaibao/goatools (accessed on December 2020)) and KEGG Orthology-Based Annotation System (http://kobas.cbi.pku.edu.cn/download.php (accessed on December 2020)).

### 4.6. Real-Time PCR

Selecting ten DEGs for real-time polymerase chain reaction (RT-PCR) to confirm the authenticity of the transcriptome results. These genes were annotated in pathways of photosynthesis, plant hormone signal transduction, phenylpropanoid biosynthesis, WRKY transcription factors, oxidation-reduction processes, plant–pathogen interactions, and alpha-linolenic acid metabolism. The primers for DEGs were designed and synthesised by Majorbio (Shanghai, China). Synthesis of cDNA was performed on HiScript Q RT SuperMix for qPCR (+gDNA wiper). RT-PCR was performed using the ABI7500 Real-time PCR System (Applied Biosystems, Waltham, MA, USA) with SYBR Premix Ex Taq II (Tli RNaseH Plus) and ROX plus (Takara Bio, Kusatsu, Japan). Expression levels of each gene were normalised relative to that of S0, and the fold-change in expression was calculated using the 2^−ΔΔCT^ method.

### 4.7. Statistical Analysis

The physiological data obtained in the experiment were processed using the Statistical Package for Social Science (SPSS; SPSS Inc., Chicago, IL, USA) version 17.0, with statistical significance set at *p* < 0.05. The chart was reprocessed using Origin version 2019 (OriginLab, Northampton, MA, USA).

## 5. Conclusions

In the two broomcorn millet varieties, NF stimulated more gene expression in response to *Sporisorium destruens* stress. KEGG enrichment analysis showed that plant hormone signal transduction, plant–pathogen interaction, and photosynthesis are important pathways in the interaction of broomcorn millet and *Sporisorium destruens*. After inoculation, 19 DEGs in the plant hormone signal transduction pathway were enriched in the JAz family of the JA signalling pathway. In the resistant variety (BM), the expression of JAz family DEGs decreased compared with CK; however, in the susceptible variety (NF), the DEGs of the JAz family after inoculation was up-regulated. In the plant–pathogen interaction pathway, the expression of DEGs from the CaM/CML, CDPK, and CNGC families in the Ca^2+^ signalling network can cause HR, cell wall reinforcement, and stomatal closure—adaptive responses to pathogen invasion. Finally, RT-PCR verification of 10 DEGs selected from the RNA-Seq transcriptome results showed that the data were reliable.

## Figures and Tables

**Figure 1 ijms-22-09542-f001:**
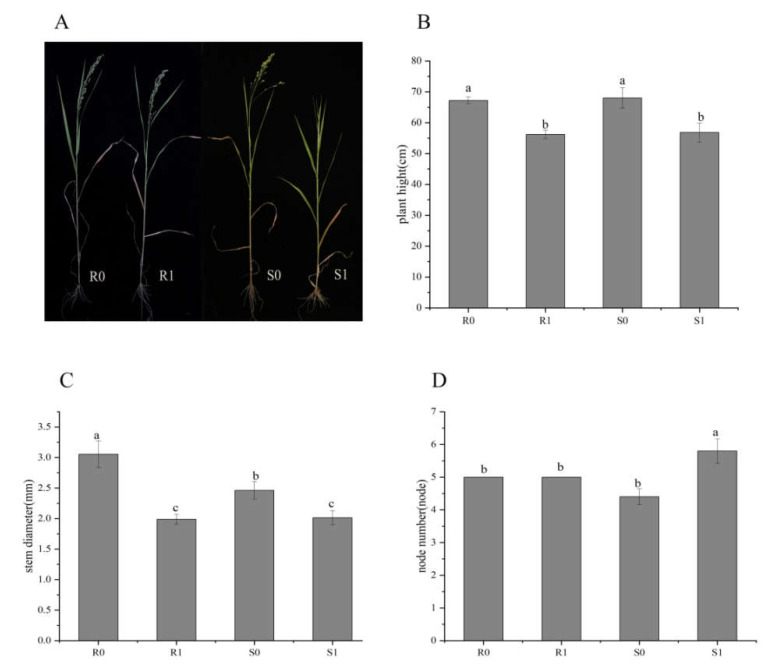
Phenotypic differences between four broomcorn millet samples: R0 and S0 were uninoculated controls; R1 and S1 were inoculated with *Sporisorium destruens*. (**A**) Phenotype at heading stage; (**B**) Plant height (cm); (**C**) Stem diameter (mm); (**D**) Node number. The values are presented as means (*n* = 5) with standard deviations. Values with different letters within the same figure were significantly different (*p* < 0.05). Data represents mean ± SD and different letters (a, b, and c) indicate significant differences (*p* < 0.05) in each samples.

**Figure 2 ijms-22-09542-f002:**
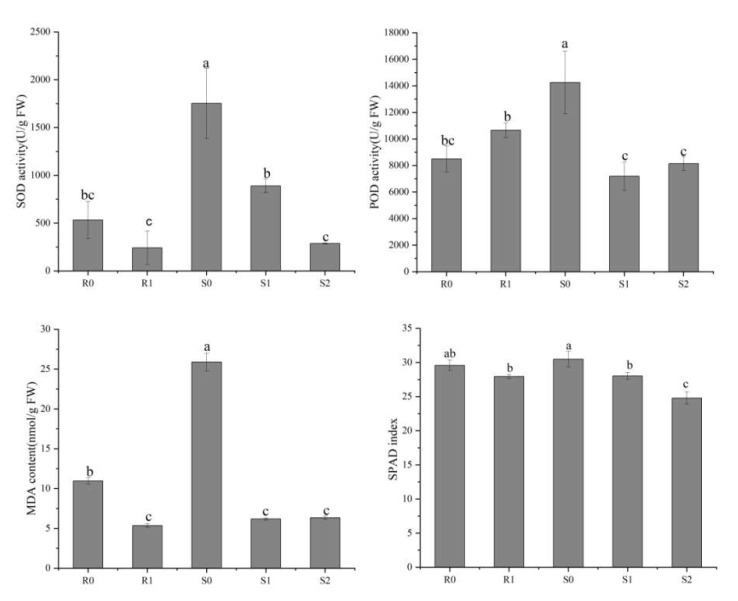
Physiological differences between four broomcorn millet samples. These physiological indices include superoxide dismutase (SOD) activity, peroxidase (POD) activity, malondialdehyde (MDA) content and SPAD index. The values are presented as means (*n* = 3) with standard deviations. Values with different letters within the same figure were significantly different (*p* < 0.05). Data represents mean ± SD and different letters (a, b, and c) indicate significant differences (*p* < 0.05) in each samples, while ab, bc indicate not significant differences (*p* > 0.05).

**Figure 3 ijms-22-09542-f003:**
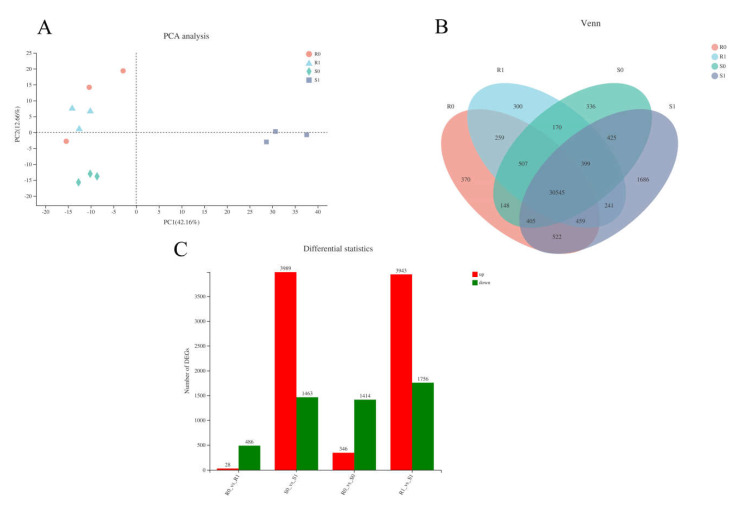
Analysis of transcriptional differences in four samples. (**A**) PCA analysis of four samples: the closer the sample, the higher the similarity. (**B**) Venn diagram of expressed genes of four samples. (**C**) The number of differentially expression genes (DEGs) in the different group.

**Figure 4 ijms-22-09542-f004:**
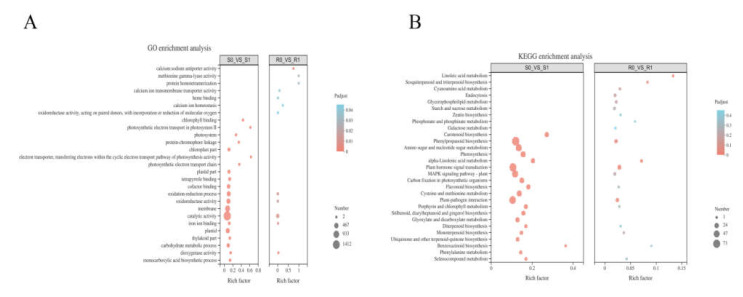
(**A**) GO enrichment analyses of the differentially expressed genes(DEGs) in NF (R0 vs. R1) and BM (S0 vs. S1) after the inoculation of *Sporisorium destruens*. (**B**) KEGG enrichment analysis of DEGs after the inoculation of *Sporisorium destruens* in R0 vs. R1 and S0 vs. S1.

**Figure 5 ijms-22-09542-f005:**
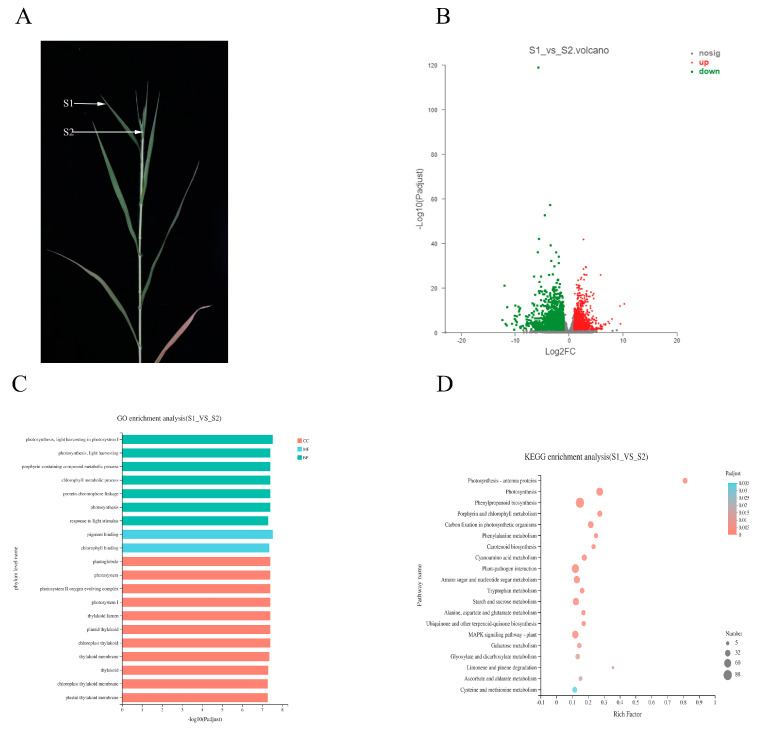
Difference between S1 and S2 in diseased plant. (**A**) The performance of S1 and S2 and their position on the diseased plant. (**B**) Distribution of DEGs in S2 compared to S1. (**C**) GO enrichment analysis of DEGs in S1 vs. S2. (**D**) KEGG enrichment analysis of DEGs in S1 vs. S2.

**Figure 6 ijms-22-09542-f006:**
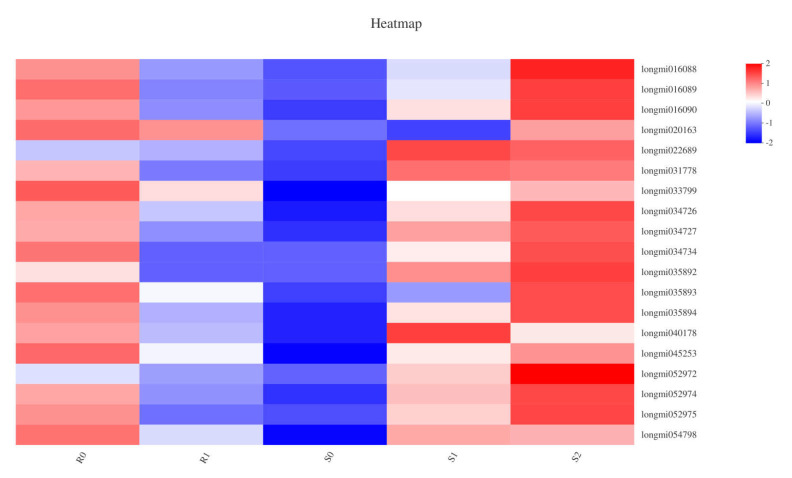
The heatmap about the correlation between DEGs enriched in JAz family and samples.

**Figure 7 ijms-22-09542-f007:**
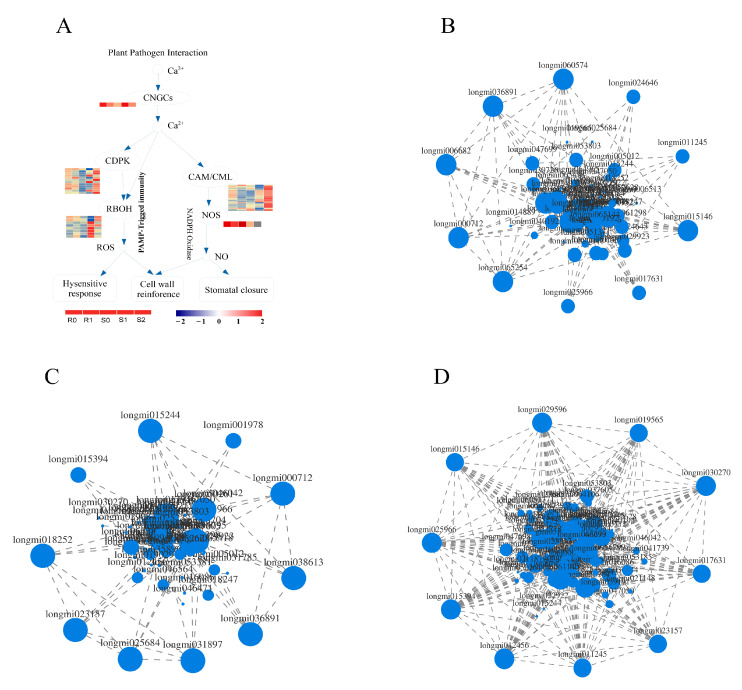
DEGs enriched in plant–pathogen interaction pathway. (**A**) DEGs related to Ca^2+^ in plant–pathogen interaction pathway. Expression related network diagram of DEGs enriched in plant–pathogen interaction pathway in R0 vs. R1 (**B**), S0 vs. S1 (**C**), and S1 vs. S2 (**D**).

**Figure 8 ijms-22-09542-f008:**
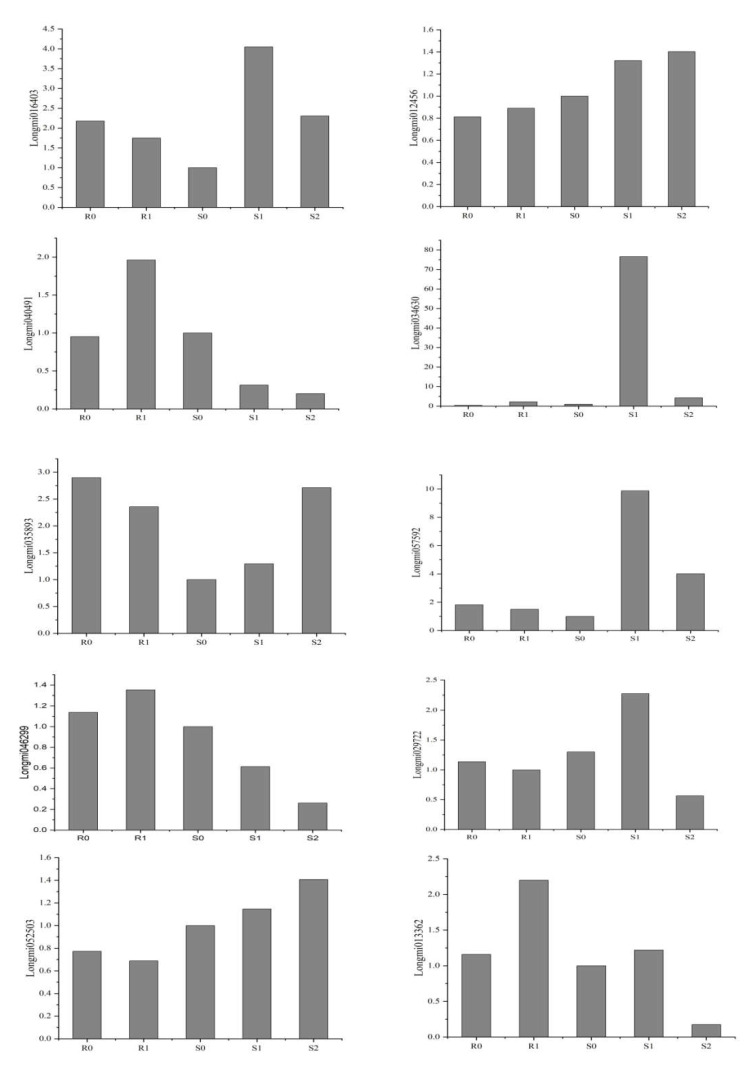
Validation of 10 genes by using RT-PCR.

## Data Availability

Not applicable.

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
