# Peer review of "Leaf Transcriptome Analysis of Broomcorn Millet Uncovers Key Genes and Pathways in Response to Sporisorium destruens"

_ijms, 2021, doi:10.3390/ijms22179542_

Round 1

Reviewer 1 Report

Jin et al. conducted an above-ground phenotypic evaluation and a transcriptomic analysis of infected leaves in two Panicum miliaceum varieties with a differential sensitivity to Sporisorium destruens. The study is very interesting, and the obtained data are valuable. Before publication, the following issues ought to be resolved.

(1) The whole manuscript needs to be corrected by a native speaker. There are excessive language errors throughout the text, including sentences without verbs as well as verbs in erroneous tense.

(2) Please start the abstract with lines 12–16 and then proceed with the aim of the study (lines 8–12). Note that the latter part (lines 8–12) includes sentences without verbs, and sentences with verbs in erroneous tense.

(3) Lines 18, 19, 43: you cannot use abbreviations without introducing them first. This issue is apparent and consistent throughout the manuscript. Please check each abbreviation in the text, whether or not this is introduced in the appropriate place.

(4) Line 88: which substrate?

(5) Line 91: please provide the color of the pot (black or white)? Why did you employ 10 plants per pot?

(6) Line 93: light intensity (not strength)

(7) Line 107: You need to provide a motivation explaining why you measured these two enzymes specifically. A high antioxidant capacity to scavenge the toxic reactive oxygen species (ROS) has been linked to stress tolerance (Chen et al. 2021, Postharvest Biol Technol 172, 111376). Antioxidant defense includes specific enzymes (SOD and POD).

(8) Line 108: You need to explain what this is, and why you measured it. Malondialdehyde (MDA) content was taken as an indication of lipid peroxidation level (Hassanvand et al. 2019, Industrial Crops Prod. 134, 19–25).

(9) Line 143: you cannot start a sentence with a number

(10) Please include the figures within the manuscript, and not as additional files.

(11) Lines 165–167: Please provide quantitative results. What about varietal differences?

(12) Line 349: regarding stomatal closure please check the review of Fanourakis et al. 2020 (Plant Physiology Biochem 153, 92–105).

(13) Line 365: what about the remaining hormones (e.g. ABA, ethylene)? These were not important, or authors did not investigate them (see Line 72)?

(14) Line 387: is low temperature relevant here?

(15) Lines 25, 393, 396, 401: authors did not measure photosynthesis, but approximated chlorophyll content (through SPAD). Therefore, more caution ought to be exercised when referring to photosynthesis per se.

(16) Line 26 (conclusion of the abstract): was this aim finally realized and by which means? In other words, do the results provide information for breeding of the species under study or provide the theoretical basis on its interaction with the disease (referred to Line 73)?  

(17) In the Introduction (Lines 43, 47) the authors introduce the two potential defense strategies (PTI, ETI). However, these two were not evaluated, and relevant results are not referred. Notably, the obtained results rather refer to functional domains of the studied genes (Line 354). Thus, consider presenting the domains in the introduction in place of PTI/ ETI.

(18) Line 93: what about the watering of the plants? How long did you cultivate the plants before inoculation and how long before sampling?

(19) Line 112: SPAD units approximate chlorophyll content

(20) Line 161: early growth stage is how many days after planting?

(21) Line 167: how do the authors explain that node (leaf) number increased following disease incidence? This is probably erroneous.  

Reviewer 2 Report

Dear Authors,

unfortunately the corresponding Figures (fig 1 -8) of the manuscript were not included in the final merged pdf file.

Therefore I was not able to review your work with the material provided. Please re-submit it including all the files.

At the current stage my only comment is that I would suggest extensive editing of the English language used. Please come in contact with a Native English speaker or relative companies that provide such type of service.

Looking forward for the full submission from the authors.

Round 2

Reviewer 1 Report

My comments were adequately addressed by the authors. Please check the language thoroughly again. Below I give some examples of language errors that need to be corrected.

Line 18: “were reduced”, “were increased”

Lines 19–20: thus all these 4 traits were decreased. So why mentioning the verb twice in a sentence?

Line 27: “lay” in place of “lays”

Line 36: 5–10% (thus the unit not mentioned twice)

Line 43: “does”

Line 44: “is”

Lines 51–54: brake this large sentence into two smaller ones

Lines 56–58: brake this large sentence into two smaller ones

Line 80: pathogen in italics